# Towards Consistent Performance on Atari using Expert Demonstrations

## Abstract

Despite significant advances in the field of deep Reinforcement Learning (RL), today's algorithms still fail to learn human-level policies consistently over a set of diverse tasks such as Atari 2600 games. We identify three key challenges that any algorithm needs to master in order to perform well on all games: processing diverse reward distributions, reasoning over long time horizons, and exploring efficiently. In this paper, we propose an algorithm that addresses each of these challenges and is able to learn human-level policies on nearly all Atari games. A new transformed Bellman operator allows our algorithm to process rewards of varying densities and scales; an auxiliary *temporal consistency loss* allows us to train stably using a discount factor of $\gamma = 0.999$ (instead of $\gamma = 0.99$) extending the effective planning horizon by an order of magnitude; and we ease the exploration problem by using human demonstrations that guide the agent towards rewarding states. When tested on a set of 42 Atari games, our algorithm exceeds the performance of an average human on 40 games using a common set of hyper parameters.

## 1 Introduction

In recent years, significant advances in the field of deep Reinforcement Learning (RL) have led to artificial agents that are able to reach human-level control on a wide array of tasks such as some Atari 2600 games (Bellemare et al., 2015). In many of the Atari games, these agents learn control policies that far exceed the capabilities of an average human player (Gruslys et al., 2018; Hessel et al., 2018; Horgan et al., 2018). However, learning human-level policies consistently across the entire set of games remains an open problem.

We argue that an algorithm needs to overcome three key challenges in order to perform well on all Atari games. The first challenge is processing diverse reward distributions. An algorithm must learn stably regardless of reward density and scale. Mnih et al. (2015) showed that clipping rewards to the canonical interval $[-1, 1]$ is one way to achieve stability. However, this clipping operation may change the set of optimal policies. For example, the agent no longer differentiates between striking a single pin or all ten pins in Bowling. Hence, optimizing the unaltered reward signal in a stable manner is crucial to achieving consistent performance across games. The second challenge is reasoning over long time horizons, which means the algorithm should be able to choose actions in anticipation of rewards that might be far away. For example, in Montezuma's Revenge, individual rewards might be separated by several hundred time steps. In the standard $\gamma$-discounted RL setting, this means the algorithm should be able to handle discount factors close to 1. The third and final challenge is efficient exploration of the MDP. An algorithm that explores efficiently is able to discover long trajectories with a high cumulative reward in a reasonable amount of time even if individual rewards are very sparse. While each problem has been partially addressed in the literature, none of the existing deep RL algorithms have been able to address these three challenges at once.

In this paper, we propose a new Deep Q-Network (DQN) (Mnih et al., 2015) style algorithm that specifically addresses these three challenges. In order to learn stably independent of the reward distribution, we use a transformed Bellman operator that reduces the variance of the action-value function. Learning with the transformed operator allows us to process the unaltered environment rewards regardless of scale and density. We prove that the optimal policy does not change in deterministic MDPs and show that under certain assumptions the operator is a contraction in stochastic

MDPs (*i.e.,* the algorithm converges to a fixed point) (see Sec. 3.2). Our algorithm learns stably even at high discount factors due to an auxiliary *temporal consistency (TC) loss*. This loss prevents the network from prematurely generalizing to unseen states (Sec. 3.3) allowing us to use a discount factor as high as $\gamma = 0.999$ in practice. This extends the effective planning horizon of our algorithm by one order of magnitude when compared to other deep RL approaches on Atari. Finally, we improve the efficiency of DQN's default exploration scheme by combining the distributed experience replay approach of Horgan et al. (2018) with the Deep Q-learning from Demonstrations (DQfD) algorithm of Hester et al. (2018). The resulting architecture is a distributed actor-learner system that combines offline expert demonstrations with online agent experiences (Sec. 3.4).

We experimentally evaluate our algorithm on a set of 42 games for which we have demonstrations from an expert human player (see Table 6). Using the same hyper parameters on all games, our algorithm exceeds the performance of an average human player on 40 games, the expert player on 34 games, and state-of-the-art agents on at least 28 games. Furthermore, we significantly advance the state-of-the-art on sparse reward games. Our algorithm completes the first level of MONTEZUMA'S REVENGE and it achieves a score of 3997 points on PITFALL! without compromising performance on dense reward games and while only using 5 demonstration trajectories.

## 2 RELATED WORK

**Reinforcement Learning with Expert Demonstrations (RLED):** RLED seeks to use expert demonstrations to guide the exploration process in difficult RL problems. Some early works in this area (Atkeson & Schaal, 1997; Schaal, 1997) used expert demonstrations to find a good initial policy before fine-tuning it with RL. More recent approaches have explicitly combined expert demonstrations with RL data during the learning of the policy or action-value function (Chemali & Lazaric, 2015; Kim et al., 2013; Piot et al., 2014). In these works, expert demonstrations were used to build an imitation loss function (classification-based loss) or max-margin constraints. While these algorithms worked reasonably well in small problems, they relied on handcrafted features to describe states and were not applied to large MDPs. In contrast, approaches using deep neural networks allow RLED to be explored in more challenging RL tasks such as Atari or robotics. In particular, our work builds upon DQfD (Hester et al., 2018), which used a separate replay buffer for expert demonstrations, and minimized the sum of a temporal difference loss and a supervised classification loss. Another similar approach is Replay Buffer Spiking (RBS) (Lipton et al., 2016), wherein the experience replay buffer is initialized with demonstration data, but this data is not kept for the full duration of the training. In robotics tasks, similar techniques have been combined with other improvements to successfully solve difficult exploration problems (Nair et al., 2017; Večerík et al., 2017).

**Deep Q-Networks (DQN):** DQN (Mnih et al., 2015) used deep neural networks as function approximators to apply RL to Atari games. Since that work, many extensions that significantly improve the algorithm's performance have been developed. For example, DQN uses a replay buffer to store off-policy experiences and the algorithm learns by sampling batches uniformly from the replay buffer; instead of using uniform samples, Schaul et al. (2015) proposed prioritized sampling where transitions are weighted by their absolute temporal difference error. This concept was further improved by Ape-X DQN (Horgan et al., 2018) which decoupled the data collection and the learning processes by having many actors feed data to a central prioritized replay buffer that an independent learner can sample from.

Durugkar & Stone (2017) observed that due to over-generalization in DQN, updates to the value of the current state also have an adverse effect on the values of the next state. This can lead to unstable learning when the discount factor is high. To counteract this effect, they constrained the TD update to be orthogonal to the direction of maximum change of the next state. However, their approach only worked on toy domains such as Cart-Pole. Finally, van Hasselt et al. (2016a) successfully extended DQN to process unclipped rewards with an algorithm called PopArt, which adaptively rescales the targets for the value network to have zero mean and unit variance.

## 3 ALGORITHM

In this section, we describe our algorithm, which consists of three components: (1) The transformed Bellman operator; (2) The temporal consistency (TC) loss; (3) Combining Ape-X DQN and DQfD.

## 3.1 DQN BACKGROUND

Let $\langle \mathcal{X}, \mathcal{A}, r, p, \gamma \rangle$ be a finite, discrete-time MDP where $\mathcal{X}$ is the state space, $\mathcal{A}$ the action space, $r$ the reward function which represents the one-step reward distribution $r(x, a)$ of doing action $a$ in state $x$, $\gamma \in [0, 1]$ the discount factor and $p$ a stochastic kernel modelling the one-step Markovian dynamics ($p(x'|x, a)$ is the probability of transitioning to state $x'$ by choosing action $a$ in state $x$). The quality of a policy $\pi$ is determined by the action-value function

$$Q^\pi : \mathcal{X} \times \mathcal{A} \to \mathbb{R}, (x, a) \mapsto \mathbb{E}^\pi \left[ \sum_{t \geq 0} \gamma^t r(x_t, a_t) \mid x_0 = x, a_0 = a \right],$$

where $\mathbb{E}^\pi$ is the expectation over the distribution of the admissible trajectories $(x_0, a_0, x_1, a_1, \dots)$ obtained by executing the policy $\pi$ starting from state $x$ and taking action $a$. The goal is to find a policy $\pi^*$ that maximizes the state-value $V^\pi(x) := \max_{a \in \mathcal{A}} Q^\pi(x, a)$ for all states $x$, *i.e.,* find $\pi^*$ such that $V^{\pi^*}(x) = \sup_\pi V^\pi(x)$ for all $x \in \mathcal{X}$. While there may be several optimal policies, they all share a common optimal action-value function $Q^*$ (Puterman, 1994). Furthermore, acting greedily with respect to the optimal action-value function $Q^*$ yields an optimal policy. In addition, $Q^*$ is the unique fixed point of the *Bellman optimality operator* $\mathcal{T}$ defined as

$$(\mathcal{T}Q)(x, a) := \mathbb{E}_{x' \sim p(\cdot|x,a)} \left[ r(x, a) + \gamma \max_{a' \in \mathcal{A}} Q(x', a') \right], \quad \forall(x, a) \in \mathcal{X} \times \mathcal{A}$$

for any $Q : \mathcal{X} \times \mathcal{A} \to \mathbb{R}$. Because $\mathcal{T}$ is a $\gamma$-contraction, we can learn $Q^*$ using a fixed point iteration. Starting with an arbitrary function $Q^{(0)}$ and then iterating $Q^{(k)} := \mathcal{T}Q^{(k-1)}$ for $k \in \mathbb{N}$ generates a sequence of functions that converges to $Q^*$.

DQN (Mnih et al., 2015) is an online-RL algorithm using a deep neural network $f_\theta$ with parameters $\theta$ as a function approximator of the optimal action-value function $Q^*$. The algorithm starts with a random initialization of the network weights $\theta^{(0)}$ and then iterates

$$\theta^{(k)} := \arg\min_\theta \mathbb{E}_{x,a} \left[ \mathcal{L}(f_\theta(x, a) - (\mathcal{T}f_{\theta^{(k-1)}})(x, a)) \right], \tag{1}$$

where the expectation is taken with respect to a random sample of states and actions from an experience replay buffer and $\mathcal{L}$ is the Huber loss (Huber, 1964) defined as

$$\mathcal{L} : \mathbb{R} \to \mathbb{R}, x \mapsto \begin{cases} \frac{1}{2}x^2 & \text{if } |x| \leq 1 \\ |x| - \frac{1}{2} & \text{otherwise} \end{cases}$$

In practice, the minimization problem in (1) is only approximately solved by performing a finite and fixed number of stochastic gradient descent (SGD) steps[1] and all expectations are approximated by sample averages.

## 3.2 TRANSFORMED BELLMAN OPERATOR

Mnih et al. (2015) have empirically observed that the errors induced by the limited network capacity, the approximate finite-time solution to (1), and the stochasticity of the optimization problem can cause the algorithm to diverge if the variance of the action-value function is too high. In order to reduce the variance, they clip the reward distribution to the interval $[-1, 1]$. While this achieves the desired goal of stabilizing the algorithm, it significantly changes the set of optimal policies. For example, consider a simplified version of BOWLING where an episode only consists of a single throw. If the original reward is the number of hit pins and the rewards were clipped, any policy that hits at least a single pin would be optimal under the clipped reward function. Instead of reducing the magnitude of the rewards, we propose to focus on the action-value function instead. We use a function $h : \mathbb{R} \to \mathbb{R}$ that reduces the scale of the action-value function. Our new operator $\mathcal{T}_h$ is defined as

$$(\mathcal{T}_h Q)(x, a) := \mathbb{E}_{x' \sim p(\cdot|x,a)} \left[ h \left( r(x, a) + \gamma \max_{a' \in \mathcal{A}} h^{-1}(Q(x', a')) \right) \right], \quad \forall(x, a) \in \mathcal{X} \times \mathcal{A}.$$

**Proposition 3.1.** *Let $Q^*$ be the fixed point of $\mathcal{T}$ and $Q : \mathcal{X} \times \mathcal{A} \to \mathbb{R}$, then*

---

[1]Mnih et al. (2015) refer to the number of SGD iterations as *target update period*.

*(i) If $h(z) = \alpha z$ for $\alpha > 0$, then $\mathcal{T}_h^k Q \xrightarrow{k \to \infty} h \circ Q^* = \alpha Q^*$.*

*(ii) If $h$ is strictly monotonically increasing and the MDP is deterministic (i.e., $p(\cdot|x,a)$ and $r(x,a)$ are point measures for all $x, a \in \mathcal{X} \times \mathcal{A}$), then $\mathcal{T}_h^k Q \xrightarrow{k \to \infty} h \circ Q^*$.*

Proposition 3.1 shows that in the basic cases when either $h$ is linear or the MDP is deterministic, $\mathcal{T}_h$ has the unique fixed point $h \circ Q^*$. We present a proof in Sec. B in the appendix. Furthermore, the fixed point iteration $\mathcal{T}_h^k Q$ converges to $h \circ Q^*$ for all $Q$. We treat the case of stochastic MDPs in the appendix (see Proposition C.1). The following proposition (see a proof in Sec. B in the appendix) shows that contracting $h$ can achieve the desired goal of decreasing the scale and variance of the action-value function.

**Proposition 3.2.** *Let $h : \mathbb{R} \to \mathbb{R}$ be a contraction mapping with fixed point 0 (i.e., $h(0) = 0$), then*

1. *$|(h \circ Q^*)(x,a)| \leq |Q^*(x,a)|$ for all $x, a \in \mathcal{X} \times \mathcal{A}$.*

2. *$\mathrm{Var}((h \circ Q^*)(X, A)) \leq \mathrm{Var}(Q^*(X, A))$ for all random variables $X, A$ on $\mathcal{X} \times \mathcal{A}$.*

In our algorithm, we use $h : z \mapsto \mathrm{sign}(z)(\sqrt{|z| + 1} - 1) + \varepsilon z$ with $\varepsilon = 10^{-2}$ where the additive regularization term $\varepsilon z$ ensures that $h^{-1}$ is Lipschitz continuous (see Proposition C.1). We chose this function because it is an invertible contraction mapping with fixed point 0 and has a closed-form inverse (see Proposition C.2).

In practice, DQN minimizes the problem in (1) by sampling transitions of the form $t = (x, a, r', x')$ from a replay buffer where $x \in \mathcal{X}, a \sim \pi(\cdot|x), r' \sim r(x,a)$, and $x' \sim p(x,a)$. Let $t_1, ..., t_N$ be $N$ transitions from the buffer with normalized sampling weights $w_1, ..., w_N$, then for $k \in \mathbb{N}$ the loss function in (1) using the operator $\mathcal{T}_h$ is approximated as

$$\mathbb{E}_{x,a}\left[\mathcal{L}(f_\theta(x,a) - (\mathcal{T}_h f_{\theta^{(k-1)}})(x,a))\right] \approx \sum_{i=1}^{N} w_i \mathcal{L}\left(f_\theta(x_i, a_i) - h(r'_i + \gamma h^{-1}(f_{\theta^{(k-1)}}(x'_i, a'_i)))\right)$$

$$=: L_{\mathrm{TD}}(\theta; (t_i)_{i=1}^N, (w_i)_{i=1}^N, \theta^{(k-1)})$$

where $a'_i := \arg\max_{a \in \mathcal{A}} f_{\theta^{(k-1)}}(x'_i, a)$ for DQN and $a'_i := \arg\max_{a \in \mathcal{A}} f_\theta(x'_i, a)$ for Double DQN (van Hasselt et al., 2016b).

### 3.3 TEMPORAL CONSISTENCY (TC) LOSS

The stability of DQN, which minimizes the TD-loss $L_{\mathrm{TD}}$, is primarily determined by the target $\mathcal{T}_h f_{\theta^{(k-1)}}$. While the transformed Bellman operator provides an atemporal reduction of the target's scale and variance, instability can still occur as the discount factor $\gamma$ approaches 1. Increasing the discount factor decreases the temporal difference in value between non-rewarding states. In particular, unwanted generalization of the neural network $f_\theta$ to the next state $x'$ (due to the similarity of temporally adjacent target values) can result in catastrophic TD backups. We resolve the problem by adding an auxiliary *temporal consistency (TC) loss* of the form

$$L_{\mathrm{TC}}(\theta; (t_i)_{i=1}^N, (w_i)_{i=1}^N, \theta^{(k-1)}) := \sum_{i=1}^{N} w_i \mathcal{L}\left(f_\theta(x'_i, a'_i) - f_{\theta^{(k-1)}}(x'_i, a'_i)\right)$$

where $k \in \mathbb{N}$ is the current iteration. The TC-loss penalizes weight updates that change the next action-value estimate $f_\theta(x', a')$. This makes sure that the updated estimates adhere to the operator $\mathcal{T}_h$ and thus are consistent over time.

### 3.4 APE-X DQFD

In this section, we describe how we combine the transformed Bellman operator and the TC loss with the DQfD algorithm (Hester et al., 2018) and distributed prioritized experience replay (Horgan et al., 2018). The resulting algorithm, which we call *Ape-X DQfD* following Horgan et al. (2018), is a distributed DQN algorithm with expert demonstrations that is robust to the reward distribution and can learn at discount factors an order of magnitude higher than what was possible before (*i.e.,* $\gamma = 0.999$ instead of $\gamma = 0.99$). Our algorithm consists of three components: (1) replay buffers; (2) actor

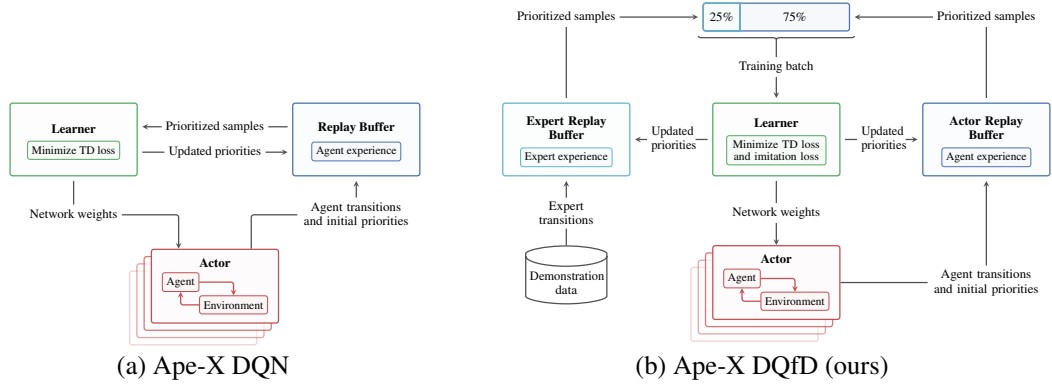

Figure 1: The figure compares our architecture (b) to the one proposed by Horgan et al. (2018) (a).

processes; and (3) a learner process. Fig. 1 shows how our architecture compares to the one used by Horgan et al. (2018).

**Replay buffers.** Following Hester et al. (2018), we maintain two replay buffers: an *actor replay buffer* and an *expert replay buffer*. Both buffers store 1-step and 10-step transitions and are prioritized (Schaul et al., 2015). The transitions in the actor replay buffer come from actor processes that interact with the MDP. In order to limit the memory consumption of the actor replay buffer, we regularly remove transitions in a FIFO-manner. The expert replay buffer is filled once offline before training commences.

**Actor processes.** Horgan et al. (2018) showed that we can significantly improve the performance of DQN with prioritized replay buffers by having many actor processes. We follow their approach and use $m = 128$ actor processes. Each actor $i$ follows an $\varepsilon_i$-greedy policy based on the current estimate of the action-value function. The noise levels $\varepsilon_i$ are chosen as $\varepsilon_i := 0.1^{\alpha_i + 3(1-\alpha_i)}$ where $\alpha_i := \frac{i-1}{m-1}$. Notably, this exploration is closer to the one used by Hester et al. (2018) and is much lower (*i.e.,* less random exploration) than the schedule used by Horgan et al. (2018).

**Learner process.** The learner process samples experiences from the two replay buffers and minimizes a loss in order to approximate the optimal action-value function. Following Hester et al. (2018), we combine the TD-loss $L_{\text{TD}}$ with a supervised imitation loss. Let $t_1, ..., t_N$ be transitions of the form $t_i = (x_i, a_i, r_i', x_i', e_i)$ with normalized sampling weights $w_1, ..., w_N$ where $e_i$ is 1 if the transition is part of the best (*i.e.,* highest episode return) expert episode and 0 otherwise. The imitation loss is a max-margin loss of the form

$$L_{\text{IM}}(\theta; (t_i)_{i=1}^N, (w_i)_{i=1}^N, \theta^{(k-1)}) := \sum_{i=1}^N w_i e_i \left( \max_{a \in \mathcal{A}}[f_\theta(x_i, a) + \lambda \delta_{a \neq a_i}] - f_\theta(x_i, a_i) \right) \quad (2)$$

where $\lambda \in \mathbb{R}$ is the *margin* and $\delta_{a \neq a_i}$ is 1 if $a \neq a_i$ and 0 otherwise. Combining the imitation loss with the TD loss and the TC loss yields the total loss formulation

$$L(\theta; (t_i)_{i=1}^N, (w_i)_{i=1}^N, \theta^{(k-1)}) := (L_{\text{TD}} + L_{\text{TC}} + L_{\text{IM}})(\theta; (t_i)_{i=1}^N, (w_i)_{i=1}^N, \theta^{(k-1)}).$$

Algo. 1, provided in the appendix, shows the entire learner procedure. Note that while we only apply the imitation loss $L_{\text{IM}}$ on the best expert trajectory, we still use all expert trajectories for the other two losses.

Our learning algorithm differs from the one used by Hester et al. (2018) in three important ways. First, we do not have a pre-training phase where we minimize $L$ only using expert transitions. We learn with a mix of actor and expert transitions from the beginning. Second, we maintain a fixed ratio of actor and expert transitions. For each SGD step, our training batch consists of 75% agent transitions and 25% expert transitions. The ratio is constant throughout the entire learning process. Finally, we only apply the imitation loss $L_{\text{IM}}$ to the best expert episode instead of all episodes.

| Algorithm | Rainbow | DQfD | Ape-X DQN | Ape-X DQfD | Ape-X DQfD (deeper) | Random | Avg. Human | Best Expert Trajectory |
|---|---|---|---|---|---|---|---|---|
| Rainbow DQN | – | 31 / 42 | 9 / 42 | 10 / 42 | 7 / 42 | 41 / 42 | 32 / 42 | 24 / 42 |
| DQfD | 11 / 42 | – | 7 / 42 | 11 / 42 | 2 / 42 | 40 / 42 | 25 / 42 | 13 / 42 |
| Ape-X DQN | 34 / 42 | 35 / 42 | – | 28 / 42 | **15 / 42** | 40 / 42 | 35 / 42 | 31 / 42 |
| Ape-X DQfD | 32 / 42 | 39 / 42 | 15 / 42 | – | 9 / 42 | 40 / 42 | 39 / 42 | 32 / 42 |
| Ape-X DQfD (deeper) | **36 / 42** | **40 / 42** | **28 / 42** | **33 / 42** | – | **42 / 42** | **40 / 42** | **34 / 42** |

Table 1: The table shows on which fraction of the tested games one approach performs at least as well as the other. The scores used for the comparison are using the no-op starts regime. As described in Sec. 4.2, we compare the agents' scores to the scores obtained by an average human player and an expert player. Ape-X DQfD (deeper) out-performs the average human on 40 of 42 games.

| | No-op starts | | | | Human starts | | | |
|---|---|---|---|---|---|---|---|---|
| | Mean | | Median | | Mean | | Median | |
| Algorithm | 42 Games | 57 Games | 42 Games | 57 Games | 42 Games | 57 Games | 42 Games | 57 Games |
| Rainbow DQN | 1022% | 874% | 231% | 231% | 897% | 776% | 159% | 153% |
| DQfD | 364% | – | 113% | – | – | – | – | – |
| Ape-X DQN | 1770% | 1695% | 421% | 434% | 1651% | 1591% | 354% | 358% |
| Ape-X DQN* (c, 0.99) | 1091% | – | 314% | – | 971% | – | 269% | – |
| Ape-X DQN* (u, 0.99) | 866% | – | 315% | – | 785% | – | 200% | – |
| Ape-X DQN* (c, 0.999) | 1546% | – | 357% | – | 1423% | – | 240% | – |
| Ape-X DQN* (u, 0.999) | 1215% | – | 280% | – | 1085% | – | 239% | – |
| Ape-X DQfD | 1536% | – | 339% | – | 1461% | – | 302% | – |
| Ape-X DQfD (deeper) | **2346%** | – | **702%** | – | **2028%** | – | **547%** | – |

Table 2: The table shows the human-normalized performance of our algorithm and the baselines. For each game, we normalize the score as $\frac{\text{alg. score} - \text{random score}}{\text{avg. human score} - \text{random score}} \times 100$ and then aggregate over all games (mean or median). Because we only have demonstrations on 42 out of the 57 games, we report the performances on 42 games and also 57 games for baselines not using demonstrations. Results of Ape-X DQN using our hyper parameters are marked with an asterisk*. The experiment Ape-X DQN* (u, 0.999) uses the exact same hyper parameters and network architecture used for the Ape-X DQfD experiments.

# 4 EXPERIMENTAL EVALUATION

We evaluate our approach on the same subset of 42 games from the Arcade Learning Environment (ALE) (Bellemare et al., 2015) used by Hester et al. (2018). We report the performance using the *no-op starts* and the *human starts* test regimes (Mnih et al., 2015).

## 4.1 HYPER PARAMETERS

Our hyper parameter setting deviates from the one used by Horgan et al. (2018) in several ways (see Tab. 3 in the appendix). As described in Sec. 3, we use a higher discount factor than Ape-X DQN (Horgan et al., 2018) ($\gamma = 0.999$ instead of $\gamma = 0.99$) and we do not clip the environment rewards to $[-1, 1]$. These changes are motivated by our goal of finding an algorithm that learns consistently on all games. In order to distinguish the contribution of changed hyper parameters from the algorithmic contributions, we rerun Ape-X DQN using variations of our hyper parameters (column Ape-X DQfD in Tab. 3). We use the naming strategy Ape-X DQN* ([c|u], [0.99|0.999]) when reporting the results where [c|u] indicates the type of reward processing (clipped or unclipped) and [0.99|0.999] is the discount factor. Tab. 2 and Fig. 6 shows the results.

We can draw two conclusions regarding reward clipping in Ape-X DQN. First, the overall performance as measured by the mean and median scores decreases when using the unclipped rewards. This shows that simplifying the reward structure of the MDPs helps Ape-X DQN learn better policies. However, when aiming at consistency (*i.e.,* having one algorithm perform well on all games), reward clipping is the wrong thing to do. When looking at our introductory example BOWLING (Fig. 6), we see that indeed reward clipping hurts performance and Ape-X DQN is able to learn a good policy only when it sees the true environment rewards. As the following experiments show, using the transformed

Bellman operator can help recover some of the performance losses incurred by using unclipped rewards.

## 4.2 BENCHMARK RESULTS

We compare our approach to Ape-X DQN (Horgan et al., 2018), on which our actor-learner architecture is based, DQfD (Hester et al., 2018), which introduced the expert replay buffer and the imitation loss, and Rainbow DQN (Hessel et al., 2018), which combines all major DQN extensions from the literature into a single algorithm. Note that the scores reported in (Horgan et al., 2018) were obtained by running 360 actors. Due to resource constraints, we limit the number of actors to 128 for all Ape-X DQfD experiments. Besides comparing our performance to other RL agents, we are also interested in comparing our scores to a human player. Because our demonstrations were gathered from an expert player, the expert scores are mostly better than the level of human performance reported in the literature (Mnih et al., 2015; Wang et al., 2016). Hence, we treat the historical human scores as the performance of an average human and the scores of our expert as expert performance.

We first analyse the performance of Ape-X DQfD with the standard dueling DQN architecture (Wang et al., 2016) that is also used by the baselines. We report the scores as *Ape-X DQfD* in Tables 1 and 2. We designed the algorithm to achieve higher consistency over a broad range of games and the scores shown in Table 1 reflect that goal. Whereas previous approaches outperformed an average human on at most 35 out of 42 games, Ape-X DQfD with the standard dueling architecture achieves a new state-of-the-art result of 39 out of 42 games. This means we significantly improve the performance on the tails of the distribution of scores over the games. When looking at this performance in the context of the median human-normalized scores reported in Table 2, we see that we significantly increase the set of games where we learn good policies at the expense of achieving lower peak scores on some games.

One of the significant changes in our experimental setup is moving from a discount factor of $\gamma = 0.99$ to $\gamma = 0.999$. Jiang et al. (2015) argue that this increases the complexity of the learning problem and, thus, requires a bigger hypothesis space. Hence, in addition to the standard architecture, we also evaluated a slightly wider (*i.e.,* double the number of convolutional kernels) and deeper (one extra fully connected layer) network architecture (see Fig. 9). With the deeper architecture, our algorithm outperforms an average human on 40 out of 42 games. Furthermore, it is the first deep RL algorithm to learn non-trivial policies on all games including sparse reward games such as MONTEZUMA'S REVENGE, PRIVATE EYE, and PITFALL!. For example, we achieve 3997 points in PITFALL!, which is below the 6464 points of an average human but far above any baseline. Finally, with a median human-normalized score of 702% and exceeding every baseline on at least $\frac{2}{3}$ of the games, we demonstrate strong peak performance and consistency over the entire benchmark.

## 4.3 IMITATION VS. INSPIRATION

Although we use demonstration data, the goal of RLED algorithms is still to learn an optimal policy that maximizes the expected $\gamma$-discounted return. While Table 1 shows that we exceed the best expert episode on 34 games using the deeper architecture, it is hard to grasp the qualitative differences between the expert policies and our algorithm's policies. In order to qualitatively compare the agent and the expert, we provide videos in the appendix (see Sec. G) and we plot the cumulative episode return of the best expert and agent episodes in Fig. 2. We see that our algorithm (——) finds more time-efficient policies than the expert (——) in all cases. This is a strong indicator that our algorithm does not do pure imitation but improves upon the demonstrated policies.

## 4.4 ABLATION STUDY

We evaluate the performance contributions of the three key ingredients of Ape-X DQfD (transformed Bellman operator, the TC-loss, and demonstration data) by performing an ablation study on a subset of 6 games. We chose sparse-reward games (MONTEZUMA'S REVENGE, PRIVATE EYE), dense-reward games (MS. PACMAN, SEAQUEST), and games where DQfD performs well (HERO, KANGAROO) (see Fig. 3).

**Transformed Bellman operator (——):** When using the standard Bellman operator $\mathcal{T}$ instead of the transformed one, Ape-X DQfD is stable but the performance is significantly worse.

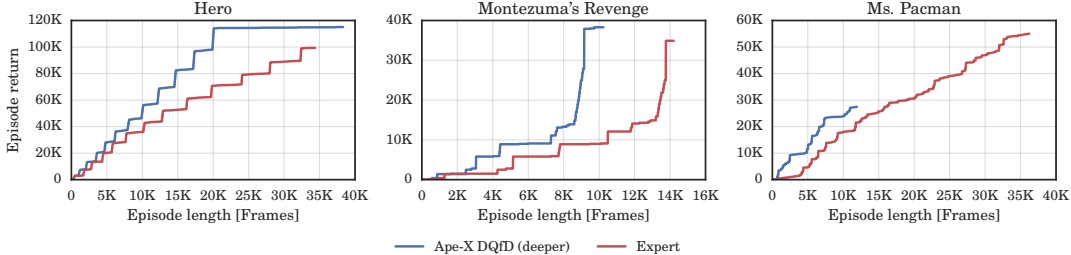

Figure 2: The figure shows the cumulative undiscounted episode return over time and compares the best expert episode to the best Ape-X DQfD episode on three games. On HERO, the algorithm exceeds the expert's performance, on MONTEZUMA'S REVENGE, it matches the expert's score but reaches it faster, and on MS. PACMAN, the expert is still superior.

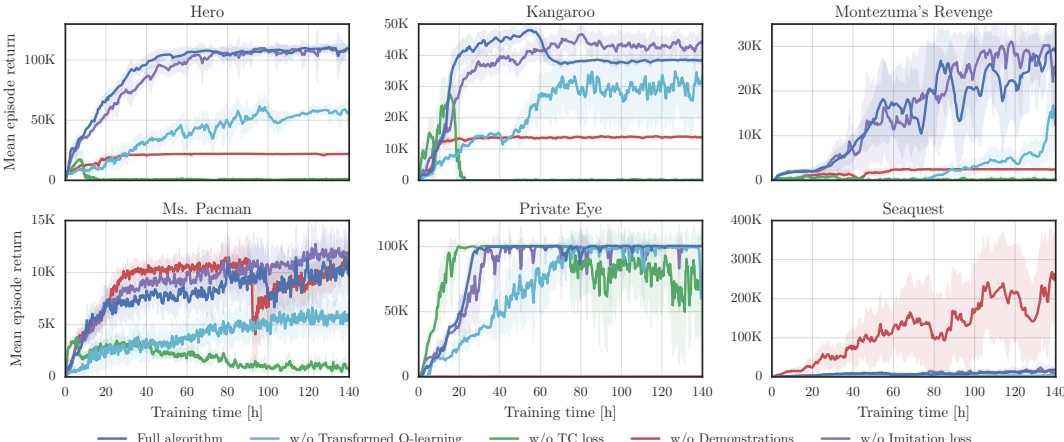

Figure 3: Results of our ablation study using the standard network architecture. The experiment without expert data (——) was performed with the higher exploration schedule used in (Horgan et al., 2018).

**TC loss (——):** In our setup, the TC loss is paramount to learning stably. We see that without the TC loss the algorithm learns faster at the beginning of the training process. However, at some point during training, the performance collapses and often the process dies with floating point exceptions.

**Expert demonstrations (—— and ——):** Unsurprisingly, removing demonstrations entirely (——) severely degrades the algorithm's performance on sparse reward games. However, in games that an $\varepsilon$-greedy policy can efficiently explore, such as SEAQUEST, the performance is on par or worse. Hence, the bias induced by the expert data is beneficial in some games and harmful in others. Just removing the imitation loss $L_{\mathrm{IM}}$ (——) does not have a significant effect on the algorithm's performance. This stands in contrast to the original DQfD algorithm and is most likely because we only apply the loss on a single expert trajectory.

### 4.5 COMPARISON TO RELATED WORK

The problems of handling diverse reward distributions and network over-generalization in deep RL have been partially addressed in the literature (see Sec. 2). Specifically, van Hasselt et al. (2016a) proposed PopArt and Durugkar & Stone (2017) used constrained TD updates. We evaluate the performance of our algorithm when using alternative solutions and report the results in Fig. 4.

**PopArt (——):** We use the standard Bellman operator $\mathcal{T}$ in combination with PopArt, which adaptively normalizes the targets in (1) to have zero mean and unit variance. While the modified algorithm manages to learn in some games, the overall performance is significantly worse than Ape-X DQfD. One possible limiting factor that makes PopArt a bad choice for our framework is that training batches contain highly rewarding states from the very beginning of training. SGD updates performed before

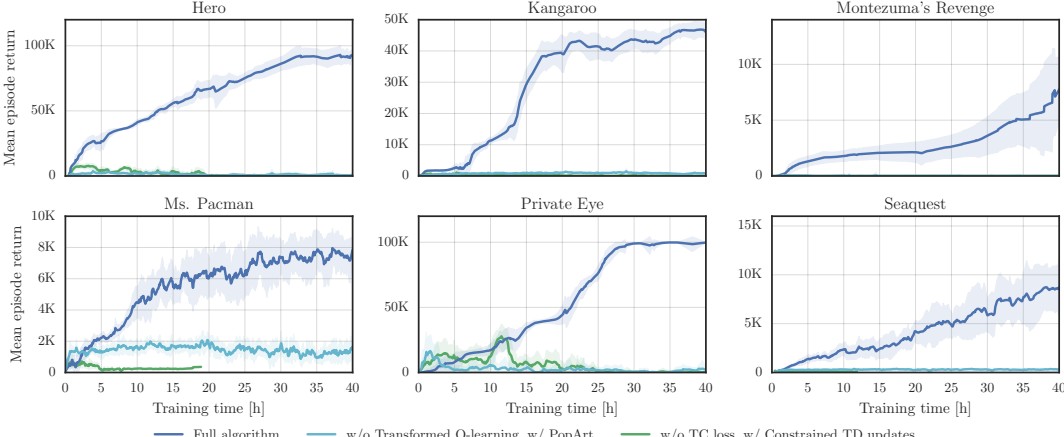

Figure 4: The figures show how our algorithm compares when we substitute the transformed Bellman operator to PopArt and when we substitute the TC loss to constrained TD updates. Note that the scales differ from the ones in Fig. 3 because the experiments only ran for 40 hours.

the moving statistics have adequately adapted the moments of the target distribution might result in catastrophic changes to the network's weights.

**Constrained TD updates (——):** We replaced the TC-loss with the constrained TD update approach (Durugkar & Stone, 2017) that removes the target network and constrains the gradient to prevent an SGD update from changing the predictions at the next state. We did not find the approach to work in our case.

## 5 CONCLUSION

In this paper, we presented a deep Reinforcement Learning (RL) algorithm that achieves human-level performance on a wide variety of MDPs on the Atari 2600 benchmark. It does so by addressing three challenges: handling diverse reward distributions, acting over longer time horizons, and efficiently exploring on sparse reward tasks. We introduce novel approaches for each of these challenges: a transformed Bellman operator, a temporal consistency loss, and a distributed RLED framework for learning from human demonstrations and task reward. Our algorithm exceeds the performance of an average human on 40 out of 42 Atari 2600 games.

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

## A    Hyperparameter comparison

| Parameter | Ape-X DQN | Ape-X DQfD | Why? |
|---|---|---|---|
| End episodes on loss of life | true | false | The demonstration trajectories recorded by the expert were recorded without the loss-of-life signal. |
| $n$-step transition parameter | $n = 3$ | $n = 1$ and $n = 10$ | $n = 1$ and $n = 10$ were the choices in the original DQfD paper. |
| Exploration of $i$-th actor where $m$ is the total number of actors and $\alpha_i := \frac{i-1}{m-1}$. | $\varepsilon_i := 0.1^{\alpha_i + 3(1-\alpha_i)}$ | $\varepsilon_i := 0.4^{\alpha_i + 7(1-\alpha_i)}$ | DQfD used low exploration noise ($\varepsilon = 0.01$) because the demonstrations help explore the MDP. |
| Number of actors | 360 | 128 | We use fewer actors in order to reduce resource consumption. |
| Optimizer | RMSProp | ADAM | We found ADAM to yield slightly better performance than RMSProp. |
| Discount factor | $\gamma = 0.99$ | $\gamma = 0.999$ | A high discount factor is crucial to solving hard-exploration games. |
| Reward processing | Clip to $[-1, 1]$ | None | With clipped rewards, games such as BOWLING cannot be solved. |

Table 3: The table compares the hyper parameters of Ape-X DQN (Horgan et al., 2018) and Ape-X DQfD. In addition to highlighting the differences, we explain the reason behind the change.

## B    Proofs from the main text

*Proof of Proposition 3.1.* (i) is equivalent to linearly scaling the reward $r$ by a constant $\alpha > 0$, which implies the proposition. For (ii) let $Q^*$ be the fixed point of $\mathcal{T}$ and note that $h \circ Q^* = h \circ \mathcal{T}Q^* = h \circ \mathcal{T}(h^{-1} \circ h \circ Q^*) = \mathcal{T}_h(h \circ Q^*)$ where the last equality only holds if the MDP is deterministic.    □

*Proof of Proposition 3.2.* Let $L_h \leq 1$ be the Lipschitz constant of $h$. (i) $|(h \circ Q^*)(x, a)| = |(h \circ Q^*)(x, a) - 0| = |(h \circ Q^*)(x, a) - h(0)| \leq L_h|Q^*(x, a) - 0| \leq |Q^*(x, a)|$. (ii) Let $X', A'$ be independent copies of $X, A$. It holds $\text{Var}((h \circ Q^*)(X, A)) = \frac{1}{2}\mathbb{E}_{X, X', A, A'}[|(h \circ Q^*)(X, A) - (h \circ Q^*)(X', A')|^2] \leq L_h^2 \mathbb{E}_{X, X', A, A'}[|Q^*(X, A) - Q^*(X,' A')|] \leq \text{Var}(Q^*(X, A))$.    □

## C    Transformed Bellman Operator in Stochastic MDPs

The following proposition shows that transformed Bellman operator is still a contraction for small $\gamma$ if we assume a stochastic MDP and a more generic choice of $h$. However, the fixed point might not be $h \circ Q^*$.

**Proposition C.1.** *Let $h$ be strictly monotonically increasing, Lipschitz continuous with Lipschitz constant $L_h$, and have a Lipschitz continuous inverse $h^{-1}$ with Lipschitz constant $L_{h^{-1}}$. For $\gamma < \frac{1}{L_h L_{h^{-1}}}$, $\mathcal{T}_h$ is a contraction.*

*Proof.* Let $Q, U : \mathcal{X} \times \mathcal{A} \to \mathbb{R}$ be arbitrary. It holds

$$\|\mathcal{T}_h Q - \mathcal{T}_h U\|_\infty = \max_{x,a \in \mathcal{X} \times \mathcal{A}} \left| \mathbb{E}_{x' \sim p(\cdot|x,a)} \left[ h \left( r(x,a) + \gamma \max_{a' \in \mathcal{A}} h^{-1}(Q(x', a')) \right) \right. \right.$$
$$\left. \left. - h \left( r(x,a) + \gamma \max_{a' \in \mathcal{A}} h^{-1}(Q(x', a')) - \right) \right] \right|$$
$$\overset{(1)}{\leq} L_h \gamma \max_{x,a \in \mathcal{X} \times \mathcal{A}} \mathbb{E}_{x' \sim p(\cdot|x,a)} \left[ \left| \max_{a' \in \mathcal{A}} h^{-1}(Q(x', a')) - \max_{a' \in \mathcal{A}} h^{-1}(U(x', a')) \right| \right]$$
$$\leq L_h \gamma \max_{x,a \in \mathcal{X} \times \mathcal{A}} \mathbb{E}_{x' \sim p(\cdot|x,a)} \left[ \max_{a' \in \mathcal{A}} \left| h^{-1}(Q(x', a')) - h^{-1}(U(x', a')) \right| \right]$$
$$\overset{(2)}{\leq} L_h L_{h^{-1}} \gamma \max_{x,a \in \mathcal{X} \times \mathcal{A}} \mathbb{E}_{x' \sim p(\cdot|x,a)} \left[ \max_{a' \in \mathcal{A}} |Q(x', a') - U(x', a')| \right]$$
$$\leq \underbrace{L_h L_{h^{-1}} \gamma}_{<1} \|Q - U\|_\infty < \|Q - U\|_\infty$$

where we used Jensen's inequality in (1) and the Lipschitz properties of $h$ and $h^{-1}$ in (1) and (2). $\qquad \square$

For our algorithm, we use $h : \mathbb{R} \to \mathbb{R}, x \mapsto \text{sign}(x)(\sqrt{|x| + 1} - 1) + \varepsilon x$ with $\varepsilon = 10^{-2}$. While Proposition C.2 shows that the transformed operator is a contraction, the discount factor $\gamma$ we use in practice is higher than $\frac{1}{L_h L_{h^{-1}}}$. We leave a deeper investigation of the contraction properties of $\mathcal{T}_h$ in stochastic MDPs for future work.

**Proposition C.2.** *Let $\varepsilon > 0$ and $h : \mathbb{R} \to \mathbb{R}, x \mapsto \text{sign}(x)(\sqrt{|x| + 1} - 1) + \varepsilon x$. It holds*

(i) *$h$ is strictly monotonically increasing.*

(ii) *$h$ is Lipschitz continuous with Lipschitz constant $L_h = \frac{1}{2} + \varepsilon$.*

(iii) *$h$ is invertible with $h^{-1} : \mathbb{R} \to \mathbb{R}, x \mapsto \text{sign}(x) \left( \left( \frac{\sqrt{1 + 4\varepsilon(|x| + 1 + \varepsilon)} - 1}{2\varepsilon} \right)^2 - 1 \right)$.*

(iv) *$h^{-1}$ is strictly monotonically increasing.*

(v) *$h^{-1}$ is Lipschitz continuous with Lipschitz constant $L_{h^{-1}} = \frac{1}{\varepsilon}$.*

We use the following Lemmas in order to prove Proposition C.2.

**Lemma C.1.** *$h : \mathbb{R} \to \mathbb{R}, x \mapsto \text{sign}(x)(\sqrt{|x| + 1} - 1) + \varepsilon x$ is differentiable everywhere with derivative $\frac{d}{dx} h(x) = \frac{1}{2\sqrt{|x| + 1}} + \varepsilon$ for all $x \in \mathbb{R}$.*

*Proof of Lemma C.1.* For $x > 0$, $h$ is differentiable as a composition of differentiable functions with $\frac{d}{dx} h(x) = \frac{1}{2\sqrt{x + 1}} + \varepsilon$. Analogously, $h$ is differentiable for $x < 0$ with $\frac{d}{dx} h(x) = \frac{1}{2\sqrt{-x + 1}} + \varepsilon$. For $x = 0$, we find

$$\lim_{z \to 0^+} \frac{h(x + z) - \overbrace{h(x)}^{=0}}{z} = \lim_{z \to 0^+} \frac{\sqrt{z + 1} - 1 + \varepsilon z}{z} \overset{\text{l'Hospital}}{=} \lim_{z \to 0^+} \overbrace{\frac{1}{2\sqrt{z + 1}}}^{\to \frac{1}{2}} + \varepsilon = \frac{1}{2} + \varepsilon$$

and similarly $\lim_{z \to 0^-} \frac{h(x+z) - h(x)}{z} = \frac{1}{2} + \varepsilon$. Hence, $\frac{d}{dx} h(x) = \frac{1}{2\sqrt{|x| + 1}} + \varepsilon$ for all $x \in \mathbb{R}$. $\qquad \square$

**Lemma C.2.** *$h^{-1} : \mathbb{R} \to \mathbb{R}, x \mapsto \text{sign}(x) \left( \left( \frac{\sqrt{1 + 4\varepsilon(|x| + 1 + \varepsilon)} - 1}{2\varepsilon} \right)^2 - 1 \right)$ is differentiable every-*

*where with derivative $\frac{d}{dx} h^{-1}(x) = \frac{1}{\varepsilon} \left( 1 - \frac{1}{\sqrt{1 + 4\varepsilon(|x| + 1 + \varepsilon)}} \right)$ for all $x \in \mathbb{R}$.*

*Proof of Lemma C.2.* For $x \neq 0$, $h^{-1}$ is differentiable as a composition of differentiable functions. For $x = 0$, it holds

$$\lim_{z \to 0^+} \frac{h^{-1}(x+z) - \overbrace{h^{-1}(x)}^{=0}}{z} = \lim_{z \to 0^+} \frac{1}{z} \frac{\left(\sqrt{1 + 4\varepsilon(z+1+\varepsilon)} - 1\right)^2 - 1}{4\varepsilon^2}$$

$$\stackrel{\text{l'Hospital}}{=} \lim_{z \to 0^+} \frac{1}{4\varepsilon^2} \frac{2\left(\sqrt{1 + 4\varepsilon(z+1+\varepsilon)} - 1\right)}{2\sqrt{1 + 4\varepsilon(z+1+\varepsilon)}} 4\varepsilon$$

$$= \lim_{z \to 0^+} \frac{1}{\varepsilon} \left(1 - \frac{1}{\sqrt{1 + 4\varepsilon(z+1+\varepsilon)}}\right) = \frac{1}{\frac{1}{2} + \varepsilon}.$$

and similarly $\lim_{z \to 0^-} \frac{h^{-1}(x+z) - h^{-1}(x)}{z} = \frac{1}{\frac{1}{2}+\varepsilon}$, which concludes the proof. $\square$

*Proof of Proposition C.2.* We prove all statements individually.

(i) $\frac{d}{dx} h(x) = \frac{1}{2\sqrt{|x|+1}} + \varepsilon x > 0$ for all $x \in \mathbb{R}$, which implies the proposition.

(ii) Let $x, y \in \mathbb{R}$ with $x < y$, using the mean value theorem, we find

$$|h(x) - h(y)| \leq \sup_{\xi \in (x,y)} \left|\frac{d}{dx} h(\xi)\right| |x - y| \leq \sup_{\xi \in \mathbb{R}} \left|\frac{d}{dx} h(\xi)\right| |x - y| = \underbrace{\left(\frac{1}{2} + \varepsilon\right)}_{=L_h} |x - y|.$$

(iii) (i) Implies that $h$ is invertible and simple substitution shows $h \circ h^{-1} = h^{-1} \circ h = \text{id}$.

(iv) $\frac{d}{dx} h^{-1}(x) = \frac{1}{\varepsilon} \left(1 - \frac{1}{\sqrt{1 + 4\varepsilon(|x|+1+\varepsilon)}}\right) > 0$ for all $x \in \mathbb{R}$, which implies the proposition.

(v) Let $x, y \in \mathbb{R}$ with $x < y$, using the mean value theorem, we find

$$|h^{-1}(x) - h^{-1}(y)| \leq \sup_{\xi \in (x,y)} \left|\frac{d}{dx} h^{-1}(\xi)\right| |x - y| \leq \sup_{\xi \in \mathbb{R}} \left|\frac{d}{dx} h^{-1}(\xi)\right| |x - y| = \underbrace{\frac{1}{\varepsilon}}_{=L_{h^{-1}}} |x - y|.$$

$\square$

# D  LEARNER ALGORITHM

---
**Algorithm 1** The algorithm used by the learner to estimate the action-value function.

---
$\theta^{(0)} \leftarrow$ Random sample
**for** $k = 1, 2, ...$ **do**
$\quad \theta^{(k)} \leftarrow \theta^{(k-1)}$
$\quad$ **for** $j = 1, ..., T_{\text{update}}$ **do** $\qquad\qquad\qquad\qquad \triangleright T_{\text{update}}$ is the target network update period
$\quad\quad (t_i, w_i)_{i=1}^{N} \leftarrow$ SAMPLEPRIORITIZED(N) $\qquad \triangleright$ Sample 75% agent and 25% expert transitions
$\quad\quad \theta^{(k)} \leftarrow$ ADAMSTEP($L(\theta^{(k)}; (t_i)_{i=1}^{N}, (w_i)_{i=1}^{N}, \theta^{(k-1)})$) $\qquad \triangleright$ Update the parameters using Adam
$\quad\quad (p_i)_{i=1}^{N} \leftarrow \left(\left|L_{\text{TD}}(\theta^{(k)}; t_i, 1, \theta^{(k-1)})\right|\right)_{i=1}^{N} \triangleright$ Compute the updated priorities based on the TD error
$\quad\quad$ UPDATEPRIORITIES($(t_1, w_1), ..., (t_N, w_N)$) $\qquad \triangleright$ Send the updated priorities to the replay buffers
$\quad$ **end for**
**end for**

---

# E  FULL EXPERIMENTAL RESULTS

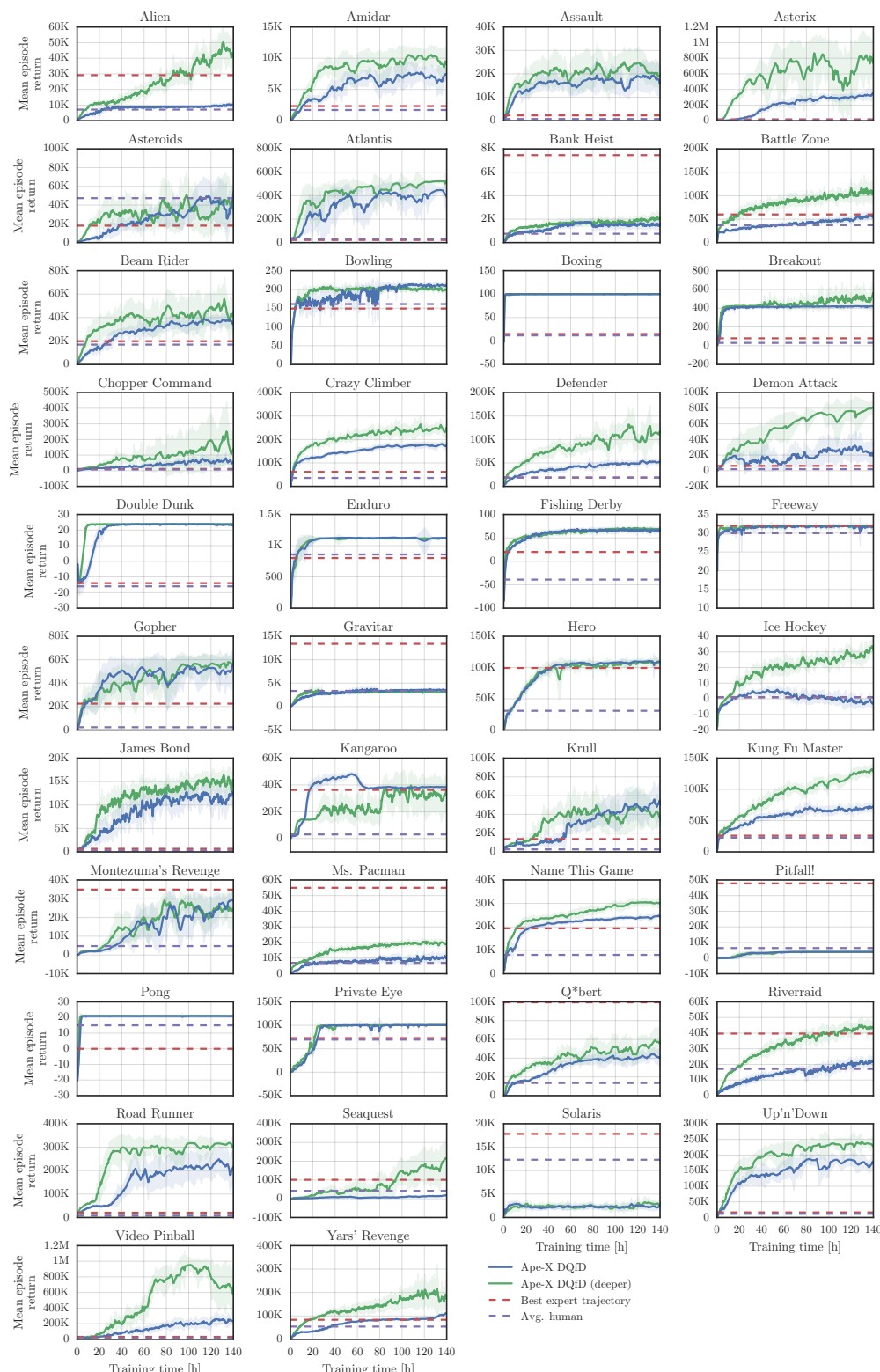

Figure 5: Training curves on all 42 games. We report the performance using the standard network architecture (Wang et al., 2016) and the slightly deeper version (see Fig. 9).

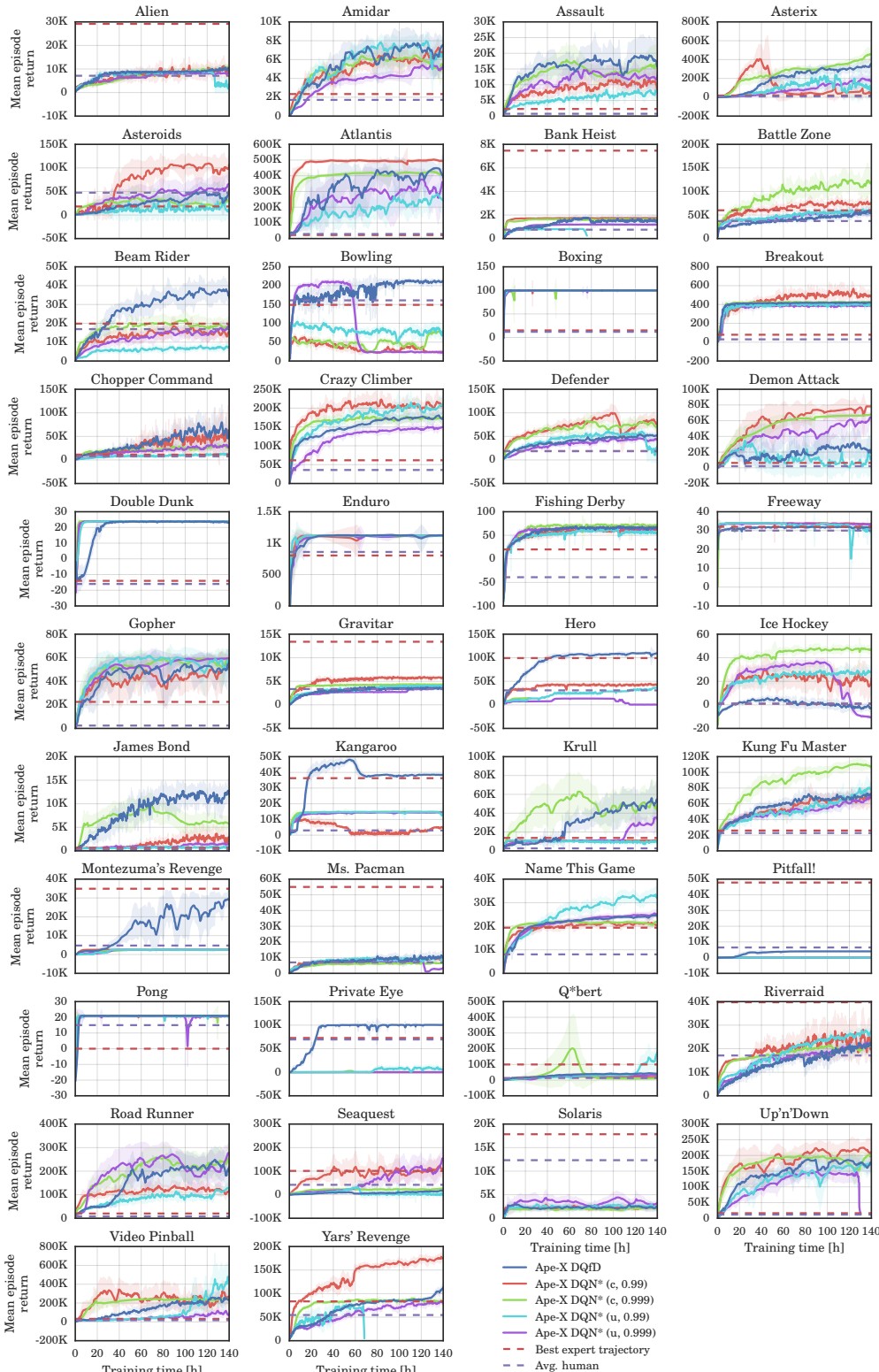

Figure 6: The curves show the effect of using clipped/unclipped rewards and low/high discount factors on all games.

| Game | Rainbow | DQfD | Ape-X DQN | Ape-X DQfD | Ape-X DQfD (deeper) | Random | Avg. Human | Expert |
|---|---|---|---|---|---|---|---|---|
| Alien | 9491.7 | 4737.5 | 40804.9 | 11313.6 | **50113.6** | 128.3 | 7128.0 | 29160.0 |
| Amidar | 5131.2 | 2325.0 | 8659.2 | 8463.8 | **12291.7** | 11.8 | 1720.0 | 2341.0 |
| Assault | 14198.5 | 1755.7 | 24559.4 | 22855.0 | **35046.9** | 166.9 | 742.0 | 2274.0 |
| Asterix | **428200.3** | 5493.6 | 313305.0 | 399888.0 | 418433.5 | 164.5 | 8503.0 | 18100.0 |
| Asteroids | 2712.8 | 3796.4 | **155495.1** | 116846.4 | 112573.6 | 871.3 | 47389.0 | 18100.0 |
| Atlantis | 826659.5 | 920213.9 | 944497.5 | 911025.0 | **1057521.0** | 13463.0 | 29028.0 | 22400.0 |
| Bank Heist | 1358.0 | 1280.2 | 1716.4 | 2061.9 | 2578.9 | 21.7 | 753.0 | **7465.0** |
| Battle Zone | 62010.0 | 41708.2 | 98895.0 | 60540.0 | **128925.0** | 3560.0 | 37188.0 | 60000.0 |
| Beam Rider | 16850.2 | 5173.3 | 63305.2 | 47129.4 | **87257.4** | 254.6 | 16926.0 | 19844.0 |
| Bowling | 30.0 | 97.0 | 17.6 | **216.3** | 210.9 | 35.2 | 161.0 | 149.0 |
| Boxing | 99.6 | 99.1 | **100.0** | **100.0** | 98.5 | -0.1 | 12.0 | 15.0 |
| Breakout | 417.5 | 308.1 | **800.9** | 419.7 | 641.9 | 1.6 | 30.0 | 79.0 |
| Chopper Command | 16654.0 | 6993.1 | 721851.0 | 96653.0 | **840023.5** | 644.0 | 7388.0 | 11300.0 |
| Crazy Climber | 168788.5 | 151909.5 | **320426.0** | 176598.5 | 247651.0 | 9337.0 | 35829.0 | 61600.0 |
| Defender | 55105.0 | 27951.5 | **411943.5** | 151442.0 | 218006.3 | 1965.5 | 18689.0 | 18700.0 |
| Demon Attack | 111185.2 | 3848.8 | 133086.4 | 100200.9 | **141444.6** | 208.3 | 1971.0 | 6190.0 |
| Double Dunk | -0.3 | -20.4 | **23.5** | 23.0 | 23.2 | -16.0 | -16.0 | -14.0 |
| Enduro | 2125.9 | 1929.8 | **2177.4** | 1663.1 | 1910.1 | 81.8 | 860.0 | 803.0 |
| Fishing Derby | 31.3 | 38.4 | 44.4 | 66.1 | **68.0** | -77.1 | -39.0 | 20.0 |
| Freeway | **34.0** | 31.4 | 33.7 | 32.0 | 31.7 | 0.1 | 30.0 | 32.0 |
| Gopher | 70354.6 | 7810.3 | **120500.9** | 114702.6 | 114168.9 | 250.0 | 2412.0 | 22520.0 |
| Gravitar | 1419.3 | 1685.1 | 1598.5 | 4214.3 | 3920.5 | 245.5 | 3351.0 | **13400.0** |
| Hero | 55887.4 | 105929.4 | 31655.9 | 112042.4 | **114248.2** | 1580.3 | 30826.0 | 99320.0 |
| Ice Hockey | 1.1 | -9.6 | **33.0** | 3.4 | 32.9 | -9.7 | 1.0 | 1.0 |
| James Bond | 19809.0 | 2095.0 | **21322.5** | 12889.0 | 16956.3 | 33.5 | 303.0 | 650.0 |
| Kangaroo | 14637.5 | 14681.5 | 1416.0 | 47676.5 | **48599.0** | 100.0 | 3035.0 | 36300.0 |
| Krull | 8741.5 | 9825.3 | 11741.4 | 104160.3 | **140670.6** | 1151.9 | 2666.0 | 13730.0 |
| Kung Fu Master | 52181.0 | 29132.0 | 97829.5 | 67957.5 | **137804.5** | 304.0 | 22736.0 | 25920.0 |
| Montezuma's Revenge | 384.0 | 4638.4 | 2500.0 | 29384.0 | 27926.5 | 25.0 | 4753.0 | **34900.0** |
| Ms. Pacman | 5380.4 | 4695.7 | 11255.2 | 12857.1 | 20872.7 | 197.8 | 6952.0 | **55021.0** |
| Name This Game | 13136.0 | 5188.3 | 25783.3 | 24465.8 | **31569.4** | 1747.8 | 8049.0 | 19380.0 |
| Pitfall! | 0.0 | 57.3 | -0.6 | 3996.7 | 3997.5 | -348.8 | 6464.0 | **47821.0** |
| Pong | 20.9 | 10.7 | 20.9 | **21.0** | 20.9 | -18.0 | 15.0 | 0.0 |
| Private Eye | 4234.0 | 42457.2 | 49.8 | **100747.4** | 100724.9 | 662.8 | 69571.0 | 72800.0 |
| Q*bert | 33817.5 | 21792.7 | **302391.3** | 71224.4 | 91603.5 | 183.0 | 13455.0 | 99450.0 |
| Riverraid | 22920.8 | 18735.4 | **63864.4** | 24147.7 | 47609.9 | 588.3 | 17118.0 | 39710.0 |
| Road Runner | 62041.0 | 50199.6 | 222234.5 | 507213.0 | **578806.5** | 200.0 | 7845.0 | 20200.0 |
| Seaquest | 15898.9 | 12361.6 | **392952.3** | 13603.8 | 318418.0 | 215.5 | 42055.0 | 101120.0 |
| Solaris | 3560.3 | 2616.8 | 2892.9 | 2529.8 | 3428.9 | 2047.2 | 12327.0 | **17840.0** |
| Up'n'Down | 125754.6 | 82555.0 | 401884.3 | 324505.2 | **469548.3** | 707.2 | 11693.0 | 16080.0 |
| Video Pinball | 533936.5 | 19123.1 | 565163.2 | 243320.1 | **922518.0** | 20452.0 | 17668.0 | 32420.0 |
| Yars' Revenge | 102557.0 | 61575.7 | 148594.8 | 109980.9 | **498947.1** | 1476.9 | 54577.0 | 83523.0 |

Table 4: Scores obtained by evaluating the best checkpoint for 200 episodes using the no-op starts regime.

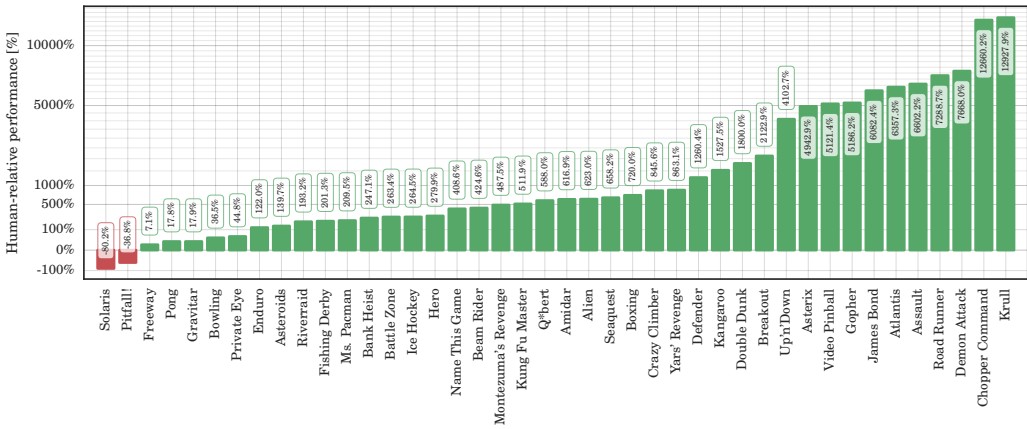

Figure 7: The human-relative score of Ape-X DQfD (deeper) using the no-ops starts regime. The score is computed as $\frac{\text{alg. score} - \text{avg. human score}}{\text{avg. human score} - \text{random score}} \times 100$.

| Game | Rainbow | DQfD | Ape-X DQN | Ape-X DQfD | Ape-X DQfD (deeper) | Random | Avg. Human | Expert |
|---|---|---|---|---|---|---|---|---|
| Alien | 6022.9 | – | **17731.5** | 1025.5 | 6983.4 | – | 6371.3 | – |
| Amidar | 202.8 | – | 1047.3 | 310.5 | 1177.5 | – | **1540.4** | – |
| Assault | 14491.7 | – | 24404.6 | 23384.3 | **34716.5** | – | 628.9 | – |
| Asterix | 280114.0 | – | 283179.5 | **327929.0** | 297533.8 | – | 7536.0 | – |
| Asteroids | 2249.4 | – | **117303.4** | 95066.6 | 95170.9 | – | 36517.3 | – |
| Atlantis | 814684.0 | – | 918714.5 | 912443.0 | **1020311.0** | – | 26575.0 | – |
| Bank Heist | 826.0 | – | 1200.8 | 1695.9 | **2020.5** | – | 644.5 | – |
| Battle Zone | 52040.0 | – | **92275.0** | 42150.0 | 74410.0 | – | 33030.0 | – |
| Beam Rider | 21768.5 | – | 72233.7 | 46454.5 | **82997.1** | – | 14961.0 | – |
| Bowling | 39.4 | – | 30.2 | **178.3** | 174.4 | – | 146.5 | – |
| Boxing | 54.9 | – | **80.9** | 64.5 | 69.7 | – | 9.6 | – |
| Breakout | 379.5 | – | **756.5** | 145.1 | 365.5 | – | 27.9 | – |
| Chopper Command | 10916.0 | – | 576601.5 | 90152.5 | **681202.5** | – | 8930.0 | – |
| Crazy Climber | 143962.0 | – | **263953.5** | 141468.0 | 196633.5 | – | 32667.0 | – |
| Defender | 47671.3 | – | **399865.3** | 37771.8 | 123734.8 | – | 14296.0 | – |
| Demon Attack | 109670.7 | – | 133002.1 | 97458.8 | **142189.0** | – | 3442.8 | – |
| Double Dunk | -0.6 | – | **22.3** | 20.5 | 21.8 | – | -14.4 | – |
| Enduro | **2061.1** | – | 2042.4 | 1538.3 | 1754.9 | – | 740.2 | – |
| Fishing Derby | 22.6 | – | 22.4 | **26.3** | 24.0 | – | 5.1 | – |
| Freeway | **29.1** | – | 29.0 | 23.8 | 26.8 | – | 25.6 | – |
| Gopher | 72595.7 | – | **121168.2** | 115654.7 | 115392.1 | – | 2311.0 | – |
| Gravitar | 567.5 | – | 662.0 | 972.0 | 1021.8 | – | **3116.0** | – |
| Hero | 50496.8 | – | 26345.3 | 104942.1 | **107144.0** | – | 25839.4 | – |
| Ice Hockey | -0.7 | – | **24.0** | 3.3 | 18.4 | – | 0.5 | – |
| James Bond | 18142.3 | – | **18992.3** | 12041.0 | 15010.0 | – | 368.5 | – |
| Kangaroo | 10841.0 | – | 577.5 | 25953.5 | **28616.0** | – | 2739.0 | – |
| Krull | 6715.5 | – | 8592.0 | 111496.1 | **122870.1** | – | 2109.1 | – |
| Kung Fu Master | 28999.8 | – | 72068.0 | 50421.5 | **102258.0** | – | 20786.8 | – |
| Montezuma's Revenge | 154.0 | – | 1079.0 | **22781.0** | 22730.5 | – | 4182.0 | – |
| Ms. Pacman | 2570.2 | – | 6135.4 | 1880.8 | 4007.4 | – | **15375.0** | – |
| Name This Game | 11686.5 | – | 23829.9 | 22874.6 | **29416.0** | – | 6796.0 | – |
| Pitfall! | -37.6 | – | -273.3 | 3367.5 | 3208.7 | – | **5998.9** | – |
| Pong | **19.0** | – | 18.7 | 14.0 | 18.6 | – | 15.5 | – |
| Private Eye | 1704.4 | – | 864.7 | 61895.1 | 54976.0 | – | **64169.1** | – |
| Q*bert | 18397.6 | – | **380152.1** | 41419.6 | 51159.3 | – | 12085.0 | – |
| Riverraid | 15608.1 | – | **49982.8** | 18720.1 | 42288.9 | – | 14382.2 | – |
| Road Runner | 54261.0 | – | 127111.5 | 486082.0 | **507490.0** | – | 6878.0 | – |
| Seaquest | 19176.0 | – | **377179.8** | 15526.1 | 269480.0 | – | 40425.8 | – |
| Solaris | 2860.7 | – | 3115.9 | 2235.6 | 1835.8 | – | **11032.6** | – |
| Up'n'Down | 92640.6 | – | **347912.2** | 200709.3 | 298361.8 | – | 9896.1 | – |
| Video Pinball | 506817.2 | – | **873988.5** | 194845.0 | 832691.1 | – | 15641.1 | – |
| Yars' Revenge | 93007.9 | – | 131701.1 | 82521.8 | **466181.8** | – | 47135.2 | – |

Table 5: Scores obtained by evaluating the best checkpoint for 200 episodes using the human starts regime.

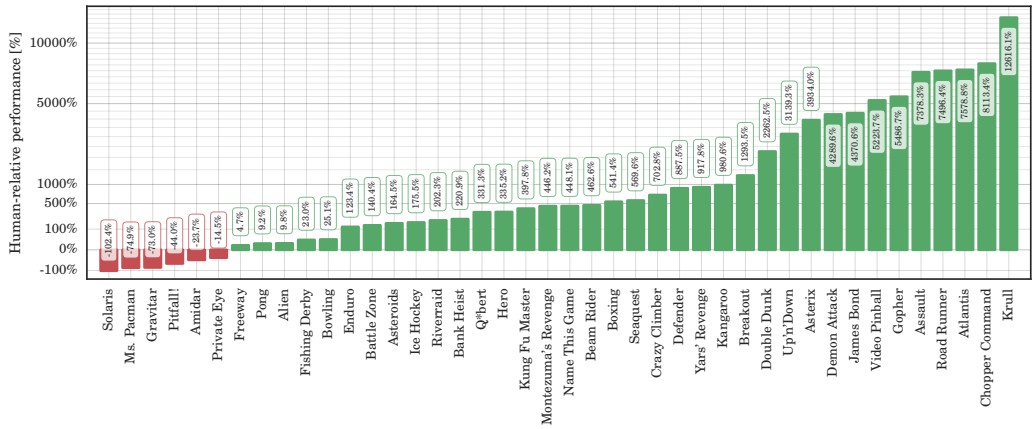

Figure 8: The human-relative score of Ape-X DQfD (deeper) using the human starts regime. The score is computed as $\frac{\text{alg. score} - \text{avg. human score}}{\text{avg. human score} - \text{random score}} \times 100$.

# F   EXPERIMENTAL SETUP & HYPER PARAMETERS

| Game | Min score | Max score | Number of transitions | Number of episodes |
|---|---|---|---|---|
| Alien | 9690 | 29160 | 19133 | 5 |
| Amidar | 1353 | 2341 | 16790 | 5 |
| Assault | 1168 | 2274 | 13224 | 5 |
| Asterix | 4500 | 18100 | 9525 | 5 |
| Asteroids | 14170 | 18100 | 22801 | 5 |
| Atlantis | 10300 | 22400 | 17516 | 12 |
| Bank Heist | 900 | 7465 | 32389 | 7 |
| Battle Zone | 35000 | 60000 | 9075 | 5 |
| Beam Rider | 12594 | 19844 | 38665 | 4 |
| Bowling | 89 | 149 | 9991 | 5 |
| Boxing | 0 | 15 | 8438 | 5 |
| Breakout | 17 | 79 | 10475 | 9 |
| Chopper Command | 4700 | 11300 | 7710 | 5 |
| Crazy Climber | 30600 | 61600 | 18937 | 5 |
| Defender | 5150 | 18700 | 6421 | 5 |
| Demon Attack | 1800 | 6190 | 17409 | 5 |
| Double Dunk | -22 | -14 | 11855 | 5 |
| Enduro | 383 | 803 | 42058 | 5 |
| Fishing Derby | -10 | 20 | 6388 | 4 |
| Freeway | 30 | 32 | 10239 | 5 |
| Gopher | 2500 | 22520 | 38632 | 5 |
| Gravitar | 2950 | 13400 | 15377 | 5 |
| Hero | 35155 | 99320 | 32907 | 5 |
| Ice Hockey | -4 | 1 | 17585 | 5 |
| James Bond | 400 | 650 | 9050 | 5 |
| Kangaroo | 12400 | 36300 | 20984 | 5 |
| Krull | 8040 | 13730 | 32581 | 5 |
| Kung Fu Master | 8300 | 25920 | 12989 | 5 |
| Montezuma's Revenge | 32300 | 34900 | 17949 | 5 |
| Ms Pacman | 31781 | 55021 | 21896 | 3 |
| Name This Game | 11350 | 19380 | 43571 | 5 |
| Pitfall | 3662 | 47821 | 35347 | 5 |
| Pong | -12 | 0 | 17719 | 3 |
| Private Eye | 70375 | 74456 | 10899 | 5 |
| Q-Bert | 80700 | 99450 | 75472 | 5 |
| River Raid | 17240 | 39710 | 46233 | 5 |
| Road Runner | 8400 | 20200 | 5574 | 5 |
| Seaquest | 56510 | 101120 | 57453 | 7 |
| Solaris | 2840 | 17840 | 28552 | 6 |
| Up N Down | 6580 | 16080 | 10421 | 4 |
| Video Pinball | 8409 | 32420 | 10051 | 5 |
| Yars' Revenge | 48361 | 83523 | 21334 | 4 |

Table 6: The table shows the performance of our expert player and the amount of available demonstrations per game. Note that the total number of episodes/trajectories is very low.

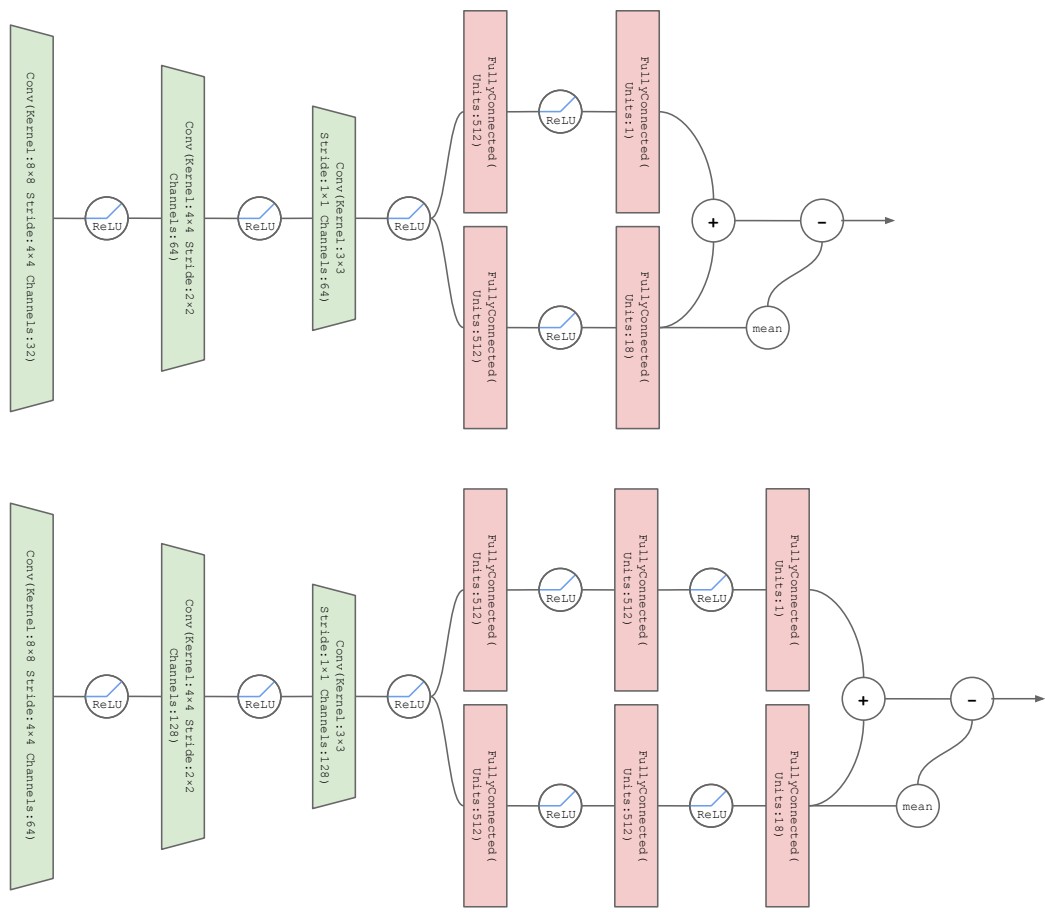

Figure 9: The two network architectures that we used. The upper one is the standard dueling architecture of Wang et al. (2016) and the lower one is a slightly wider and deeper version.

| Parameter | Comment | Value |
|---|---|---|
| | Learner configuration | |
| Batch size | | 256 |
| Agent transitions per batch | | 192 |
| Expert transitions per batch | | 64 |
| Adam learning rate | | $5 \cdot 10^{-5}$ |
| Adam regularizer | | $\frac{0.01}{256}$ |
| Maximum gradient norm | We use `tf.clip_by_global_norm` | 40.0 |
| Target update period | Referred to as $T_{\text{target}}$ in the text | 2500 |
| Discount factor | Referred to as $\gamma$ in the text | 0.999 |
| Margin | Referred to as $\lambda$ in the text | $\sqrt{0.999}$ |
| | Arcade Learning Environment (ALE) parameters | |
| Use full Atari action set | | Yes |
| Repeat actions | | 4 |
| Expose lives | | No |

Table 7: The table shows all of our hyper parameters.

## G  VIDEOS

Due to the size of the videos, we uploaded them to an anonymized Google Drive account.

**MONTEZUMA'S REVENGE:** https://drive.google.com/file/d/1oW57PKpWSYvc-KaZBG0Z-VQwn29Rkiql/view

**HERO:** https://drive.google.com/file/d/1AXfTuXvUHPDXWskZKV0-2WbzJu-dbSLL/view

**BOWLING:** https://drive.google.com/file/d/1MKeFOilKn7rX4MH-koZlHKTQINWZGybU/view

**BREAKOUT:** https://drive.google.com/file/d/1ZKce3Vva2VfXguC1m09ZEkKB_qDOJeNj/view

