# OpenReview forum: "Towards Consistent Performance on Atari using Expert Demonstrations"
_ICLR.cc/2019/Conference_

### Official Review · AnonReviewer3 · 2018-11-01
**Three loosely related methods which are all well justified**

**Rating:** 7
**Confidence:** 4

**Review:**

Summary: This paper proposes three new techniques for improving Atari performance over APE (Horgan 2018).  Two of them are closely linked in that they deal with improving stability.  Another involves integrating the use of expert trajectories from DQfD.

I will summarize each:

Transformed Bellman: This applies a rescaling function (it's basically a monotonically increasing version of the sqrt(x) function) to the Q-function and applies the inverse of the function to the max Q-value of the next state (such that the contracting effect h-function is not "applied" multiple times when doing the TD backup).

Temporal Consistency: This encourages the "next state" after where the TD-update is applied to not change too much.  This addresses a problem discussed in (Durugkar 2018).  I think the intuition here is that the state which follows the state with the TD update may be visually similar, but it does not impact the value in the past states, so its value function should not have a highly correlated change with the previous state's change in value function.

DQfD: Storing an expert replay buffer and an actor replay buffer.  The expert replay buffer is fixed and the actor replay buffer stores the most recent "actor processes".  Train with both a supervised imitation loss (only for the highest return episode) and the original TD loss.  Additionally, the pre-training phase is removed and the ratio of expert-learned trajectories is fixed (both seem like steps in the right direction).

Review: This paper proposes a few changes to DQN training, two of which are aimed at reducing instability, and one is aimed at improving exploration (expert trajectories).  Because all of these changes are well justified and the experiments are fairly thorough, I recommend acceptance.  My main reservation is that the ideas presented are not very strongly thematically linked.  The presence of ablation studies compensates for this to some extent.

Strengths:

  -The discussion of related work and comparison to baselines is pretty extensive.  For example I appreciated the ablation study removing "transformed Q-learning" and comparison to the pop-art method.

  -The results, at least for Ape-X DQfD seem impressive to me in that the method works without reward clipping and with a much higher discount factor.  Additionally the results generally outperform DQfD (uses expert trajectories) and Rainbow (no human trajectories).  Additionally evidence was presented that the learned policies often exceed the performance of the human demonstrations (for example in time to achieve rewards).

Weaknesses:

  -Two of the techniques: "transformed bellman" and "temporal consistency" seem well-linked thematically, but the expert demonstration idea seems orthogonal.  I would have preferred splitting that idea out into a separate paper, given that the paper is already 20 pages.

  -The motivation for temporal consistency just references (Durugkar 2018).  The readability of this paper would be improved if it were discussed more here as well.  I also feel like the analysis could be more thorough here, for example a result using the temporal consistency loss on Baird's counter example really should be shown (like figure 2 in Durugkar's paper).

-It would be nice to see a visualization or a toy problem with the "transformed bellman".

Questions:

-Is the "highest return episode" idea (3.4) general or is it exploiting the fact that Atari is deterministic?  It seems like in general we'd want to use many high reward episodes, or the highest reward episodes that go into different parts of state space.  It seems like it could be a very bad idea on certain settings (for example if the reward has a lot of randomness).

-"Proposition 3.1 shows that in the basic cases when either h is linear or the MDP is deterministic, Th
has the unique fixed point h ◦ Q∗".  From 3.1, it looks if h is linear, then it distributes over r(x,a) + maxh^{-1}(Q) and then it also won't effect which is the max, so it would reduce to h*r(x,a) + max(Q) - which means it's just rescaling the original reward.  So then this result is trivial?  Please correct me if I misunderstood something here.

-Could an MDP be constructed which causes the transformed bellman operator to perform badly?  I am imagining something where the MDP is just a single step, and there is a stochastic action which behaves like a lottery.  So perhaps there is a 1-in-1-million chance to win 1-billion dollars by taking an action.  If I understand correctly the transformed bellman operator will destroy the large reward here (because in a single step, there is just r(x,a) which h is applied to).  Which would make the action seem bad even though it's actually appealing.

Notes:

  -I did not read the proofs in the appendix.

---

> ### Author Response · Authors · 2018-11-12
> **Thank you for the review!**
>
> We would like to thank the reviewer for their thorough review and comments.
>
> Concerning the strengths and weaknesses highlighted by the reviewer, we agree that the three techniques employed may seem orthogonal. We choose to present them in a single paper for two reasons:
>
> First, adding expert replay can make training significantly more unstable. Consider the plots of Ms. Pacman in Fig. 3 and Fig. 6. Even when increasing the discount factor and removing clipped rewards Ape-X DQN does mostly fine (training collapses after >100h). However, when continuously replaying high-rewarding episodes (total reward ~55k points), training becomes unstable without the transformed Bellman operator and the TC-loss. Hence, side-stepping the exploration problem and continuously exposing the learning algorithm to high-rewarding trajectories amplifies the instability problems.
>
> The second reason concerns our goal with this paper. We want to improve consistency and generality. Previous algorithms either focused on hard-exploration games exclusively or only optimized the mean/median scores (where a few ill-performing games don’t hurt too much). With this paper, we show that it is possible to perform well across the board. Our algorithm achieves good scores on hard-exploration games while also performing well in the mean/median benchmark.
>
> Regarding the temporal consistency loss, we do not/cannot argue that this loss can solve Baird’s counter-example. It merely improves instability in our application by restricting the changes in network predictions (similarly, Durugkar’s constrained gradient technique does not solve Baird’s counter-example but only avoids divergence).
>
> Concerning the questions:
>
> 1.  The highest episode reward idea is indeed something specific to the deterministic nature of Atari and helps on very hard exploration games. As highlighted by the reviewer, this should not be used in a very stochastic environment.
>
> 2.  The reviewer is correct in that the first part of the proposition is trivial (the case when h is linear). However, we find the the second part is less obvious (i.e. the fixed point of the transformed operator is the transformed fixed point of the original operator). In Sec. C (appendix), we present a more general investigation of the case when h is Lipschitz and the MDP is stochastic.
>
> 3.  The definition of “badly” depends on the notion of optimality. For example, the standard Bellman operator wouldn’t score many points on Atari when the discount factor of the MDP is low. So, instead of finding MDPs where the operator performs badly, we focus on MDPs where our operator behaves differently (i.e. the learned policy is different). Proposition 3.1 implies that such an MDP needs to be stochastic and we can construct an example.
> Consider a bandit-like episodic MDP with two states s_start, s_terminal and two actions a_1, a_2. After taking an action a in s_start, a reward r(s_start, a) is received and the episode terminates in s_terminal. We can disregard the discount factor because of the unit episode length. The reward distributions are as follows:
> r(s_start, a_1) = 2
> r(s_start, a_2) = Uniform({1, 3.1})
>
> The standard Bellman operator learns the action-value function:
> Q*(s_start, a_1) = 2, Q*(s_start, a_2) = 2.05.
>
> Our operator on the other hand learns the action-value function:
> Q*(s_start, a_1) = 0.752.., Q*(s_start, a_2) = 0.740..
>
> Hence, the greedy-policy learned by the standard operator selects a_2 and the greedy-policy learned by our operator selects a_1.

---

### Official Review · AnonReviewer2 · 2018-11-08
**Educated guess**

**Rating:** 7
**Confidence:** 1

**Review:**

The paper reads well, proposes well motivated modifications to existing methods, and gets what appears to be strong results. I have no experience in RL, and although I read the paper I don't feel able to make meaningful comments.

---

### Official Review · AnonReviewer4 · 2018-11-12
**Interesting method but needs more experiments to support**

**Rating:** 5
**Confidence:** 4

**Review:**

This paper propose a method that aims to solve the following 3 problems: sensitivity to unclipped reward, robustness to the value of the discount factor, and the exploration problem.

Pros
This paper propose a transformed Bellman operator, and the author proved its convergence under some deterministic MDP conditions. The proposed transformed Bellman operator is interesting since that is analogous to some variance reduction techniques in the policy gradient literature. In the value based method literatures, those techniques have not been well studied.

Cons
I think the main issue of this paper is the experiments can not fully support the advantage claim of the proposed method.
	1. With the author's hyper-parameters, the proposed method (Ape-X DQfD) has worse performance than the baseline Ape-X DQN, with the original hyper parameter of Ape-X DQN (Table 1). The author has a version of the baseline with the same hyper parameter as the proposed method, but the modified one is worse than the original baseline, which is not satisfactory. I think in general we should try to keep the original hyper parameter especially the original performance is better.
	2. With the same hyper parameters, the performance of the proposed method Ape-X DQfD is better than Ape-X DQN*(with reward clipping, gamma=0.999) with human starts but worse with no-op starts (Table 2). On the whole, their performance I would say, is similar. That makes reader questions about the utility of not use reward clipping, since without reward clipping, we did optimize the true objective, but the final performance is sometimes better and sometimes worse. I am afraid that undo the reward clipping is making the problem unnecessarily harder.
	3. The transformed Bellman operator transforms the Q function by a contraction. It's interesting to see what kind of effect of some ad-hoc transformations on the reward will behave. Given the particular function form the author have used, it's especially interesting to see how this transformation: r' = sgn( R) sqrt(abs(r )) will affects the performance.
	4. The authors ablates the method on 6 games out of the 42. However, it's mostly qualitative, rather than quantitative. I think it would be more convincing if the leave-one-out experiment could be carried out on all 42 games.
	5. The author combines Ape-X DQN with a modified version of DQfD, as mentioned in Section 3.4. For a fair comparison, I think there should be a corresponding modified version of DQfD as a baseline.

I think the author proposed an interesting approach, however, the experiment section, especially the ablation section could be improved. It's hard to tell how much the transformed Bellman operator and the temporal consistency loss contributes on an average case, based on the current results. If the author could provide more information, I'm willing to change my review.

---

> ### Author Response · Authors · 2018-11-14
> **Thanks for the review!**
>
> We would like to thank the reviewer for their comments.
>
> 1.  The reviewer argues that “Ape-X DQfD [using our hyper parameters] has worse performance than Ape-X DQN” using the original hyper parameters. This statement is not unconditionally true and depends on the evaluation metric. We want to improve consistency and evaluate this in terms of the number of games on which our algorithm exceeds average human performance. Using this metric, Ape-X DQfD beats Ape-X DQN 39 to 35.
>
> We agree that we should have made this clearer when stating our claims. We will address this issue in the revised version.
>
> We do agree with the reviewer that changing hyper parameters should be avoided. However, when choosing a new research objective (consistency across the benchmark rather than mean performance), one might need to adjust the problem setting correspondingly. This is why we not only list the hyper parameters we have changed in Table 3, but for each hyper parameter we explain why we changed it.
>
> 2.  We disagree with the reviewer who believes that undoing reward clipping makes the problem “unnecessarily harder”. In the paper, we illustrate using the game Bowling how reward clipping makes it impossible to learn good policies (similarly, Pitfall suffers as abundant but small (~ -1) negative rewards immediately overshadow the sparse but big (~ 4000) rewards). We don’t want to just chase higher mean/median scores by simply getting better on games previous algorithms already perform super-human on. We want to build an algorithm that can solve more games without having to custom fit hyper parameters to individual games. As our examples show, such an algorithm must process unclipped rewards.
>
> The reviewer, furthermore, insists on Ape-X DQN being better than Ape-X DQfD with gamma=0.999. However, this score “Ape-X DQN* (c, 0.999)” was obtained using reward clipping. When comparing using the same hyper parameters (“Ape-X DQN* (u, 0.999)”), Ape-X DQN performs worse in all four metrics presented.
>
> 3.  Studying the “effect of some ad-hoc transformations” on rewards rather than the value function could be indeed interesting but it is not the main purpose of the paper. We may refer the reviewer to the original DQfD algorithm (Hester et al. 2017) that uses such a transformation.
>
> 4.  We agree that an ablation study on 42 games would be more interesting. However, doing the leave-one-out study on all games is too compute intensive at the moment (For the actors alone we’d need 5 ablations x 3 seeds x 42 games x 128 actors x 140h = 11,289,600 CPU hours). Choosing a subset of games for the ablation study is in line with previous work (e.g. [Horgan et al. 2018]). We carefully chose the six games to be as informative as possible (and explain how we chose them), which is why we do think that our ablation study provides sufficient information to understand the effect of each contribution.
>
> 5.  Simply adding more actors to the original DQfD algorithm is a poor choice of baseline. First, DQfD uses a single replay buffer and relies on prioritization to select expert transitions. This works fine when there is a single actor process feeding data into the replay buffer. However, in a distributed setup (with 128 actors), the ratio of expert to actor experience drops significantly and prioritization no longer exposes the learner to expert transitions. Second, DQfD uses a pre-training phase during which all actor processes are idle. This wastes a lot of compute resources when running a decent number of actors. Many of our contributions help eliminate these shortcomings.

---

### Official Review · AnonReviewer5 · 2018-11-13
**Combining three simple and orthogonal ideas achieves good results on Atari game**

**Rating:** 6
**Confidence:** 4

**Review:**

The paper proposed three extensions to DQN to improve the learning performance on Atari games. The extensions are transformed Bellman update, temporal consistency loss and expert demonstration. These three extensions together make the proposed algorithm outperform the state-of-the-art results for Atari games. However, these extensions are relatively straightforward and thus the technical contribution is lean.

The first extension, transformed Bellman update is similar to reward scaling. While reward scaling is a linear transform, this paper proposed a non-linear transform. It would be great if the paper can compare the transformed Bellman update with reward scaling: For example, normalize the reward based on the best expert performance. In addition, the proposed transformation seems a bit ad-hoc. I feel that many similar transforms will work. For example, would a log transform work? It would be impressive if this transform is learned simultaneously, instead of manually crafted.

The second extension, TC loss, is a double-edge blade. While it stabilizes the learning process, it can slow down the learning. It is not clear how this paper balances these two? How much weights are placed for the TC loss. I feel that the functionality of the TC loss is somewhat similar to the target network. Will reducing the frequency of updating the target network achieve the similar effect?

The third extension, bootstrapping from demonstration data, can greatly help the learning process. Although the paper enumerates three differences to DQFD, I am not convinced that these differences lead to significant better results.

Overall, this paper is clearly written. It achieves good results. The contributions are three orthogonal extensions to DQN, which are relatively straightforward. For above reasons, I do not feel strongly about the paper. I am OK if the paper is accepted.

---

> ### Author Response · Authors · 2018-11-16
> **Thanks for the review!**
>
> We would like to thanks the reviewer for their comments.
>
> Concerning the Bellman operator
> We experimented with linear reward scaling extensively. Because there is no universal scaling constant that stabilizes training on all games, we did try various methods of deriving a scaling factor from demonstrations (mean/max reward, mean/max discounted return, etc.). We failed to find an algorithmic way of linearly scaling the reward that works consistently on all 42 games.  Some games (e.g. Seaquest) have exponential reward structures, which are difficult to address using constant linear reward scaling. This is why we looked into non-linear transformations that - like linear transformations - preserve the optimal policy (at least in deterministic MDPs).
>
> Other transforms will work. As our theoretical analysis shows, the transform only needs to be monotonic in deterministic MDPs and h, h^-1 being Lipschitz will still preserve convergence in stochastic MDPs.
>
> Unfortunately, a smoothed logarithmic transform h(z) = sign(z) * log(|z| + 1) does not have a Lipschitz continuous inverse (h^-1(z) = sign(z) * (exp(|z|) - 1)) and when adding the eps * x term to ensure Lipschitz continuity of the inverse, we can no longer find a closed-form representation of the inverse.
>
> We would like to highlight that using DQN with unclipped rewards on Atari is long-standing problem. Previous approaches such as POPART are more complex and require careful engineering. Our transformed Bellman operator is easy to understand, easy to implement, and works reliably in practice. While the contribution might seem straightforward, it addresses a long-standing problem with a novel solution that works even when no expert data is available [see e.g. https://openreview.net/forum?id=r1lyTjAqYX ].
>
> Concerning the TC loss
> The reviewer rightfully points out the trade-off between training speed and stability. For example, the plots for Private Eye in Fig. 3 clearly show that training without the TC loss is faster (albeit potentially unstable). We don’t actively balance training speed and stability as our results show that final performance is not impacted by the TC loss. Hence, we keep experiments running a bit longer.
>
> We don’t introduce weights for the three losses (TD, TC, Imitation). The objective function L given in Sec. 3.4 is the one implemented in the code used to obtain all results.
>
> We have not experimented with longer update cycles for the target network. This is an interesting idea. We will try to run additional experiments. However, it is unlikely we will have the necessary capacity to do so before the end of the rebuttal period.
>
> Concerning DQfD
> Our changes are primarily technical and port the single-actor DQfD algorithm to a distributed setup. Rather than “leading to better results” by being novel ideas, these changes (continuous expert replay, disjoint prioritization, no pretraining) are necessary when running DQfD in a distributed setup. Please see point 5 in our response to AnonReviewer4 below to see why simply running DQfD with many actors is a poor choice.
>
> =====
> Edit: Made the link clickable.

---

### Meta-Review · Area_Chair1 · 2018-12-13
**Combination of three techniques.**

**Confidence:** 4
**Recommendation:** Reject

**Metareview:**

This paper proposes a combination of three techniques to improve the learning performance of Atari games. Good performance was shown in the paper with all three techniques together applied to DQN. However, it is hard to justify the integration of these techniques. It is also not clear why the specific decisions were made when combining them. More comprehensive experiments, such as a more systematic ablation study, are required to convince the benefits of individual components. Furthermore, it seems very hard to tell whether the improvement of existing approaches, such as Ape-X DQN, was from using the proposed techniques or a deeper architecture (Tables 1&2&4&5). Overall, this paper is not ready for publication.